# A Novel Soft Alignment Approach for Language Models with Explicit Listwise Rewards

## Abstract

Existing alignment methods, such as Direct Preference Optimization (DPO), are mainly tailored for pairwise preference data where rewards are implicitly defined rather than explicitly given. In this paper, we introduce a general framework for large language model alignment, leveraging a novel optimization objective to bridge the gap in handling reward datasets with a list of responses explicitly annotated with scalar preferences scores. Our work comprise a novel algorithm, soft preference optimization, SPO, which enables the direct extraction of an LM policy from reward data as well as preference data. The core of SPO is a novel listwise preference optimization objective with the exponential-logarithm function form and a adaptive loss coefficient that inject listwise preference signals into the large language model. We evaluate our methods in both reward and preference settings with Mistral models in different sizes. Experiments suggest that our method surpasses various preference baselines when reward datasets are available. We also find our method significantly outperforms DPO in complex reasoning tasks like math and coding.

## 1 Introduction

Aligning pretrained Language Models (LMs) with scalar rewards that reflect human intentions is crucial for enhancing their ability to follow instructions (Ouyang et al., 2022; OpenAI, 2023). These rewards can be given either explicitly or implicitly. Explicit rewards can be scalar ratings of human annotators or advanced models like GPT-4 (OpenAI, 2023), while implicit rewards are usually preference labels assigned to pairwise responses. One effective approach for aligning LMs with preference data is Direct Preference Optimization (DPO, Rafailov et al. (2024b)). DPO applies a reward training loss but parameterizes the reward model as the response likelihood ratio between two LMs, allowing for training reward models and extracting LM policies simultaneously. This approach is more streamlined and thus more favorable compared with traditional Reinforcement Learning (RL) methods Ouyang et al. (2022), which typically require a two-stage training process: first training reward models, then extracting LM policies.

Despite its simplicity and effectiveness, DPO is only tailored for preference data $(x \rightarrow \{y_w > y_l\}$ with $K = 2$ responses per instruction $x$. When multiple responses are available, directly assigning a scalar reward to each response is usually more convenient and efficient than comparing them in a pairwise manner. The resulting reward datasets $(x \rightarrow \{(y_i, r_i)\}_{1:K})$, however, cannot be directly leveraged for DPO training. Previous work Tunstall et al. (2023) usually prunes reward datasets by selecting the best response and pairing it with a random remaining one. This is suboptimal as all reward values and additional dispreferred responses are thrown away in its data-preprocessing process. In addition, a well-observed problem with DPO is that the likelihood of the preferred response tends to decrease throughout training Pal et al. (2024); Rafailov et al. (2024a). We find this issue arises mainly from DPO's focus on adjusting the relative likelihood across different responses per instruction.

To address these issues, we present Soft Preference Optimization (SPO), an alignment method that allows directly extracting LM policies from both reward datasets and preference datasets with arbitrary response numbers. Notably, SPO subsumes DPO loss as a special case under pairwise preference settings and can thus be seen as a natural extension of DPO. We show DPO is a binary classification loss while SPO is its multi-category version. However, unlike DPO which is built

upon assumptions of Bradley-Terry models, SPO derives from Information Noise Contrastive Estimation (InfoNCE, (Oord et al., 2018)) and knowledge distillation (Hinton et al., 2015). We further propose SPO-abs, an alternative alignment method to SPO, which adds a regularization term for the absolute values to mitigate the decreasing reward issue. SPO-abs differs from SPO by only loss definition and is also suitable for both preference and reward datasets.

We evaluate our methods on the Mistral-7B models from two dimensions. When reward datasets (Cui et al.) are available, we show that directly applying our reward-based alignment offers clear improvement compared with preference-based algorithms, achieving higher evaluation performance in evaluations conducted by the GPT-4 model as a judge. We further validate this improvement comes from SPO/SPO-abs's ability to fully leverage the additional sub-optimal responses. When only preference data is given (Yuan et al., 2024b), we compare the SPO-abs method against the DPO loss. Our experimental results spanning various benchmarks show that SPO-abs outperforms DPO in complex reasoning tasks such as math and coding.

Our main contributions are:

- We propose SPO, a preference optimization method based on the reward datasets. In addition, SPO-abs resolves the data likelihood decline issue of DPO. The two proposed methods are uniquely suited for both reward and preference data, offering a general framework that integrates preference-based algorithms.
- We conduct experiments on both reward datasets and preference datasets, demonstrating that our methods can outperforms various preference methods by fully exploiting data information in reward datasets.

## 2 RELATED WORKS

**Development of language models.** Recently, the development of language models has undergone significant changes (Zhao et al., 2023), evolving from initial rule-based approaches to today's data-driven deep learning models. A milestone advancement began in 2018 when Google introduced the BERT model (Devlin et al., 2019) with a Transformer architecture (Vaswani et al., 2017), which achieved remarkable results in various natural language processing tasks through unsupervised training. Following this, even larger models like GPT-3 (Brown et al., 2020), ChatGPT (Ouyang et al., 2022) and GPT-4 (OpenAI, 2023) further propelled this trend, not only reaching unprecedented scales with hundreds of billions of parameters but also demonstrating versatility and flexibility across a diverse range of NLP tasks. These advancements have spurred researchers to explore applications of LLMs in areas such as machine translation, text summarization, dialogue systems, and even multimodal understanding.

While it is important to develop large language models in the size of 100 billions, the even wider applications rely on the relatively smaller sized models. These compact models, while perhaps not matching the sheer breadth of knowledge or depth of understanding afforded by their gargantuan counterparts (Zeng, 2023), offer significant advantages in terms of deployment flexibility, computational efficiency, and accessibility across diverse platforms and devices, including those with limited resources. Consequently, efforts aimed at refining smaller models to optimize both efficacy and practicality are critical for broadening the reach and utility of NLP solutions in everyday contexts. Recently, the community has witnessed the releases of open-sourced models like LlaMA-2 7B (Touvron et al., 2023), Mistral-7B (Jiang et al., 2023). These models can run an edge device like laptop or cellphone, after being deployed under frameworks like Ollama[1].

**Alignment of large language model.** The alignment of large language models (LLMs) is critical for ensuring they operate ethically, accurately, and responsibly Shen et al. (2023). Proper alignment helps prevent the generation of harmful or biased content, promotes factual correctness and reliability, and supports legal compliance and user trust Wang et al. (2023b). Additionally, it ensures that LLMs contribute positively to society by respecting social values, enhancing educational integrity, and maintaining accessibility and inclusivity for all users. As these models become more integrated into daily life, their alignment with societal norms and expectations becomes increasingly important to foster beneficial interactions and outcomes.

---

[1]https://github.com/ollama/ollama

Current approaches cater to either explicit reward data or preference data, often lacking the versatility to address both concurrently. Reinforcement Learning (Schulman et al., 2017) is inherently suitable for explicit reward scenarios. However, its on-policy nature necessitates learning a reward model from data first, leading to an indirect two-stage optimization process (Christiano et al., 2017; Ouyang et al., 2022; Shen et al., 2024). Recent developments in preference-based alignment techniques (Rafailov et al., 2024b; Azar et al., 2024; Ethayarajh et al., 2024; Wang et al., 2023a; Hong et al., 2024) have streamlined this process. They enable direct alignment of LMs through a singular loss, but this comes at the expense of being confined to pairwise preference data. Other alignment approaches (Yuan et al., 2024a; Song et al., 2024) are also not tailored for aligning with reward datasets. Recent work (Cai et al., 2023) attempts to extend DPO's parameterization technique to explicit reward contexts. However, it only considers binary rewards. In comparison, our methods can handle both continuous rewards and preference data.

## 3 PRELIMINARIES: DIRECT PREFERENCE OPTIMIZATION

LM alignment is essentially a constrained policy optimization problem:

$$\max_{\pi_\theta} \mathbb{E}_{p(x)} \left[ \mathbb{E}_{\pi_\theta(y|x)} r(x,y) - \alpha D_{KL} \left( \pi_\theta(\cdot|x) || \mu(\cdot|x) \right) \right], \tag{1}$$

where $\mu$ represents the pretrained LM. $x$ and $y$ are respectively instructions and responses. $r$ is a reward function that reflects human intentions. $\alpha$ is the temperature coefficient. Peng et al. (2019) has proved that the optimal solution for the optimization problem in Equation 1 is:

$$\pi^*(y|x) = \mu(y|x) \frac{e^{r(x,y)/\alpha}}{Z(x)} \propto \mu(y|x) e^{r(x,y)/\alpha}. \tag{2}$$

Direct Preference Optimization (DPO) (Rafailov et al., 2024b) assumes we only have access to some pairwise preference data $x \to \{y_w > y_l\}$ for each instruction $x$. The preference probability of human annotators is modeled by a learnable implicit reward model $r_\theta$ under Bradley-Terry theories (Bradley & Terry, 1952):

$$\pi_\theta(y_w > y_l|x) := \sigma(r_\theta(y_w, x) - r_\theta(y_l, x)), \tag{3}$$

where $\sigma$ is the sigmoid function. To learn $r_\theta$, DPO simply adopts a binary classification loss:

$$\mathcal{L}_{\text{DPO}} = -\mathbb{E}_{\{x, y_w > y_l\}} \log \sigma \left( r_\theta(y_w, x) - r_\theta(y_l, x) \right). \tag{4}$$

In practice, the latent function $r_\theta$ is parameterized by the log-likelihood ratio between $\pi_\theta$ and $\mu$:

$$r_\theta(x, y) := \beta \log \frac{\pi_\theta(y|x)}{\mu(y|x)}, \tag{5}$$

where $\beta$ a linear coefficient for scaling $r_\theta$. This parameterization is crucial because it ensures $\pi^\theta(y|x) \propto \mu(y|x) e^{r_\theta(x,y)/\beta}$ constantly hold. It transforms generative policy optimization into a simple discriminative classification task: When $r_\theta = r$ and $\beta = \alpha$ are satisfied, we naturally have $\pi_\theta = \pi^*$.

## 4 METHOD

### 4.1 PREFERENCE DATA VERSUS REWARD DATA

Compared with constructing preference datasets, annotating each response with scalar rewards can be more flexible and convenient. Preference methods are only suitable for pairwise data ($x \to \{y_w > y_l\}$) and would require $C_K^2$ evaluations for comparing $K$ responses. In contrast, reward datasets ($x \to \{y_i, r_i\}_{1:K}$) allow an arbitrary number of responses per prompt with $K$ evaluations.

Despite its simplicity in handling preference data, DPO is not tailored for reward datasets. We introduce a new alignment method termed SPO to mitigate this gap. We show that reward alignment can be solved by constructing a classification problem to identify the optimal response from multiple candidates. We then demonstrate that SPO subsumes DPO as a special case and thus is a natural extension of DPO.

## 4.2 OBJECTIVE

In essence, DPO represents response rewards as LM likelihoods and constructs a binary classification task for learning the reward model. Given that there are more than two ($K > 2$) responses per prompt in reward datasets, we seek to construct a multi-class classification task for learning reward models from explicit rewards instead of preference labels. We begin by formally defining the task.

Consider a batch of $K$ responses $\{(y_i, r_i)\}_{1:K}$ for an instruction $x$. $\{y_i\}_{1:K}$ consists of one optimal response $y_v$ that is sampled from $\pi^*(y|x) \propto \mu_t(y|x)e^{r_t(x,y)/\alpha}$, and $K-1$ suboptimal noises independently sampled from $\mu_t(y|x)$. $\nu \in \{1, \ldots, K\}$ is the random index of that optimal response. Our goal is to identify which of the $K$ candidates is $y_v$, given only reward labels $r_i$ for each response. Intuitively, the response with higher rewards should have a higher probability of being the target response. That is, given the above $K$ response candidates and their respective rewards, the posterior probability for the $\nu$-th response being drawn from $\pi^*$ can be approximately expressed as:

$$p(\nu|x, \{y_i\}_{1:K}) = \frac{\exp(r_t(x, y_\nu))}{\sum_{i=1}^{K} \exp(r_t(x, y_i))}. \tag{6}$$

Note that there are many open-sourced LLMs available that has gone through extensive alignment procedures, and some of them like Qwen-2 72B Yang et al. (2024) are quite powerful. In order to draws latent knowledge from the these powerful LLMs, we now select a large open-sourced model Qwen-2 72B to serve as $\mu_t(y|x)$, and its instruction and alignment fine-tuned version, Qwen-2 72B Instruct, as $\pi_t(y|x)$, and the teacher model's reward is given by $r_t(x, y) = \log \frac{\pi_t(y|x)}{\mu_t(y|x)}$. And we optimize the target model by:

$$\mathcal{L}_\theta^{\text{SPO}}(x, \{y_i, r_i\}_{i=1}^{K}) = -\sum_{i=1}^{K} \left[ \underbrace{\frac{e^{r_t(x,y_i)/\alpha}}{\sum_{j=1}^{K} e^{r_t(x,y_j)/\alpha}}}_{\text{teacher's soft labels}} \log \underbrace{\frac{e^{r_\theta(x,y_i)}}{\sum_{j=1}^{K} e^{r_\theta(x,y_j)}}}_{\text{student's prediction}} \right], \tag{7}$$

where $r_\theta(x, y)$ is given by

$$r_\theta(x, y) = \log \frac{\pi_\theta(y|x)}{\mu(y|x)} \tag{8}$$

The loss function is similar to the knowledge distillation loss from Hinton et al. (2015). Our methods transform generative modeling problems into classification tasks by contrasting multiple data points with the guidance of an existing powerful LLM.

**How does SPO work?** The SPO loss (Equation 7) can be seen as a $K$-category cross-entropy loss with the soft labels provided from a more powerful teacher LLM. The soft label represents the important dark knowledge from the teacher LLM, and is calculated by first calculating the teacher rewards and then go through a softmax operation.

## 4.3 DPO AS A SPECIAL CASE

We now demonstrate that DPO is a special case of our SPO in the asymptotic sense. If we set reward temperature $\alpha$ to be zero, the SPO objective becomes as follows:

$$\mathcal{L}_\theta^{\text{SPO}}(x, \{y_i, r_i\}_{1:K}) \approx -\sum_{i=1}^{K} [\mathbb{I}(r_i = r_{\max}) \log \frac{e^{r_\theta(x,y_i)}}{\sum_{j=1}^{K} e^{r_\theta(x,y_j)}}], \tag{9}$$

where $r_{max}$ is the largest one in $\{r_i\}_{1:K}$. Since in DPO, the response number $K$ is 2, the above equation continues to be simplified to

$$\mathcal{L}_\theta^{\text{SPO}}(x, \{y_i, r_i\}_{1:K}) = -\log \frac{e^{r_\theta(x,y_a)}}{e^{r_\theta(x,y_a)} + e^{r_\theta(x,y_b)}},$$
$$= -\log \sigma(r_\theta(x, y_a) - r_\theta(x, y_b)), \tag{10}$$

where $a$, $b$ are 1 or 2, and $r_a > r_b$.

Table 1: Comparison of SPO and SPO-abs algorithm for aligning language models. Both reward loss and pairwise preference loss are given.

| Alignment Method | SPO | SPO-abs |
|---|---|---|
| Modeling Target | $\pi^*(y\|x) \propto \mu_t(y\|x)e^{r_t(x,y)/\alpha}$ | |
| Reward dataset | $\mathbf{x} \to \{y_i, r_i\}_{1:K}$ | |
| Loss ($K > 1$) | $-\sum_{i=1}^{K} \frac{e^{r_t(x,y_i)/\alpha}}{\sum_{j=1}^{K} e^{r_t(x,y_i)/\alpha}} \log \frac{e^{r_\theta(x,y_i)}}{\sum_{j=1}^{K} e^{r_\theta(x,y_j)}}$ | $SPO + \frac{1}{K} \sum_{i=1}^{K} \sigma(-r_\theta(x,y_i))$ |
| Preference Dataset | $x \to \{y_w > y_l\}$ | |
| Loss ($K = 2$) | $-\log \sigma(r_\theta(x,y_a) - r_\theta(x,y_b))$ (DPO) | $DPO + \frac{1}{2} \sum_{i \in \{w,l\}} \sigma(-r_\theta(x,y_i))$ |
| Optimizing Target | relative value | relative and absolute value |
| $r_{\theta^*}(x, y_{best}) \geq 0$ | not guaranteed | guaranteed |

$\alpha$ indicates a trade-off between diversity and optimality. When $\alpha$ becomes larger, the soft label's distribution becomes flat, and its ability to guide the optimization of the target LLM will decrease. Meanwhile, if $\alpha$ decreases to 0, SPO only increases the likelihood for the optimal response and decreases the responses for the others, thus turning the reward dataset into a preference dataset.

### 4.4 ABSOLUTE VALUE REGULARIZATION

A well-observed issue with DPO is that the likelihood of all responses continually decrease throughout training (Pal et al., 2024; Rafailov et al., 2024a). We find SPO shares this trend due to their inherent equivalence. Decreased data likelihood is unsatisfying because it directly contradicts the maximum likelihood objective for supervised training and may eventually harm performance (Yuan et al., 2024b).

We hypothesize the main cause of this decreasing likelihood is that SPO methods only adjust relative rewards among responses, rather than optimizing their absolute value. To address this problem, we take inspiration from NCE (Gutmann & Hyvärinen, 2012), and propose SPO-abs, a variant to SPO. SPO-abs can also guarantee convergence to the optimal LM policy by directly learning the absolute reward for each response, thereby counteracting the decreasing likelihood trend. Formally,

$$\mathcal{L}_\theta^{\text{SPO-abs}}(x, \{y_i, r_i\}_{i=1}^K) = \mathcal{L}_\theta^{\text{SPO}}(x, \{y_i, r_i\}_{i=1}^K) + \frac{1}{K} \sum_{i=1}^{K} (\log \sigma(-r_\theta(x,y_i))),$$

$$= -\sum_{i=1}^{K} \left[ \frac{e^{r_t(x,y_i)/\alpha}}{\sum_{j=1}^{K} e^{r_t(x,y_j)/\alpha}} \log \frac{e^{r_\theta(x,y_i)}}{\sum_{j=1}^{K} e^{r_\theta(x,y_j)}} - \frac{1}{K} \sigma(-r_\theta(x,y_i)) \right]$$

(11)

The loss function for SPO-abs involves two forces that jointly determine the trend of increasing or decreasing $r_\theta(x,y)$. Responses with higher rewards would, in principle, attain higher likelihood after training.

**Comparison between the two loss objectives** Table 1 compares the optimization objectives from SPO and SPO-abs. SPO and SPO-abs both calibrate relative values of reward models across various responses $\{y_i\}_1^K$ for an instruction $x$. In other words, the absolute value of $r_\theta(x,y)$ is not directly constrained. This can lead to some counterintuitive behaviors. For instance, the learned reward for even the highest-reward response could decrease over time without contradicting the loss definition, as long as the reward margin keeps increasing. This could lead to poor performance or training instability. In contrast, SPO-abs specifically focuses on optimizing absolute values of the reward model. In practice, SPO-abs effectively prevents the likelihood of the preferred responses from decreasing. We find this is particularly helpful for challenging tasks like coding.

## 5 EXPERIMENTS

We mainly seek to answer two research questions through our experiments:

- If we have access to reward-annotated datasets with more than 2 responses per prompt, does SPO or SPO-abs offer empirical improvement compared with preference-based approaches that simply prune reward datasets into preference datasets?

- If only pairwise preference data is available, when should one choose SPO-abs over DPO? What benefits does SPO-abs offer? Note that SPO is exactly DPO in this setting.

## 5.1 EXPRIMENTS ON THE REWARD DATASET

**Reward datasets**    We employ the UltraFeedback (Cui et al.) dataset, an instruction-following dataset annotated by GPT-4. UltraFeedback aims at advancing the alignment of large language models (LLMs) with human preferences through the use of large-scale, high-quality, and diversified feedback datasets. Utilizing a vast dataset of approximately 64,000 prompts and 256,000 responses from various high-quality sources, UltraFeedback evaluates LLM responses on criteria such as instruction-following, truthfulness, honesty, and helpfulness. GPT-4 rates each response with a scalar reward on a scale of 0-10. Prior research indicates that these GPT-4 rewards closely align with human annotations (Zheng et al., 2023), establishing them as an efficient, cost-effective alternative to human feedback. This dataset is used to train models to better serve user needs while maintaining ethical standards.

**Evaluation datasets**    We choose the well-acknowledged and widely used GPT4-based benchmarks as follows for evaluating LLMs:

- MT-bench (Zheng et al., 2023). MT-Bench is a benchmark designed to evaluate the performance of chat assistants across multiple turns of conversation and various categories of tasks. It consists of 80 multi-turn questions covering common use cases that are challenging enough to differentiate among models. These questions are categorized into eight types of user prompts, including Writing, Roleplay, Extraction, Reasoning, Math, Coding, STEM, and Humanities/Social Science. Each category includes 10 multi-turn questions crafted carefully by experts. To assess the models' performance on MT-Bench, strong language models (LLMs) are used as judges, and the results indicate that these LLM judges, particularly GPT-4, can align well with both controlled and crowdsourced human preferences, achieving over 80% agreement, which is comparable to the level of agreement between humans. Thus, MT-Bench serves as a comprehensive tool to measure the effectiveness of chat assistants, complementing traditional benchmarks.

- AlpacaEval (Dubois et al., 2024) is an LLM-based automated evaluation metric consisting of a set of 805 instructions that reflect typical user interactions on the Alpaca web demo. Both a baseline model, typically GPT-4 turbo, and the model under evaluation generate responses for each instruction, which are then compared side-by-side by another GPT-4 turbo-based evaluator that outputs the likelihood of preferring the evaluated model's response. The metric calculates a win rate, representing the expected probability that the evaluated model's output is preferred over the baseline across these instructions. Initially designed for the Alpaca chatbot and AlpacaFarm simulator, AlpacaEval was intended to mitigate certain biases, such as presentation order, by randomizing sequences. However, other factors like length bias were not controlled for, leading to potential manipulation by AI systems. To address this, Length-Controlled AlpacaEval was introduced, which employs a regression-based method to adjust for biases by controlling for length differences, thereby improving the robustness and correlation with human evaluations.

**Evaluation metrics**    MT-Bench utilizes the GPT-4 as the judge. As is shown in Zheng et al. (2023), GPT-4 can assign a fair score between 1 to 10 to a LLM's response which is consistent with human annotators. We report the average GPT-4 score (gpt-4-score).

On the AlpacaEval dataset, we compare the LLM against the reference responses from the Davinci-003 (Brown et al., 2020) model. We utilize the GPT-4 as the judge and determine how often the LLM wins the reference responses (denoted as the win rate).

In addition, we will put the LLM aligned with the SPO method or the DPO method in the Chatbot Arena (Chiang et al., 2024), and count the ratio of winning rate against the other models.

**Baselines**    We exam the following baseline methods:

Table 2: Comparison between reward-based methods (SPO, SPO-abs) and preference-based methods (for example, DPO, IPO) in LLM alignment. We focus on the general instruction-following abilities of each method measured by GPT-4 evaluations and human preference. The highest number (excluding the Referenced models ) in each benchmark is **bolded** and the second highest is underlined.

| Method | Annotation Type | MT-bench | AlpacaEval | Win agaist DPO |
|---|---|---|---|---|
| *Referenced models* | | | | |
| GPT-4 | Reward Model | 9.18 | 93.8 | - |
| LLaMA2-chat-70b | Reward Model | 6.86 | 92.7 | - |
| Mistral-7B-instruct | SFT Data | 6.84 | 92.6 | - |
| *Baseline models* | | | | |
| Mixtral-7B-sft | SFT Data | 6.45 | 85.2 | - |
| + KTO | Preference | 7.12 | 91.9 | - |
| + IPO | Preference | 7.45 | 90.6 | - |
| + DPO (Zephyr-$\beta$) | Preference | 7.34 | 90.6 | - |
| + DPO-1vsO | Preference | 7.22 | 91.6 | 52.1 |
| + DPO-pw | Preference | 7.38 | 90.3 | 53.3 |
| *Ours* | | | | |
| + SPO | Reward | 7.55 | 91.1 | 56.8 |
| + SPO-abs | Reward | **7.64** | **91.9** | **58.6** |

- DPO (Rafailov et al., 2024b). To apply the DPO method with reward datasets in which more than two responses are annotated per instruction, we conduct the data transformation according to Zephyr (Tunstall et al., 2023). DPO selects the highest reward response and a random remaining one from UltraFeedback for each instruction. This procedure discards two additional suboptimal responses in the dataset as well as their reward information.

- DPO-1vsO. Comparing the above procedure of constructing DPO dataset, one might predict that applying the DPO to a more grained preference dataset would yield better results. To investigate this, we examined a variants of DPO that utilize all available responses in UltraFeedback by pair the highest-performing response with each of the other ones separately.

- DPO-pw. This variant utilizes all available responses in UltraFeedback by summing up all DPO loss possibilities for two out of all the responses.

- KTO (Ethayarajh et al., 2024). Using a Kahneman-Tversky model of human utility, KTO propose a novel method that directly maximizes the utility of generations instead of maximizing the log-likelihood of preferences, as current methods do.

- IPO (Azar et al., 2024). IPO proposes a new general objective for learning from human preferences that is expressed in terms of pairwise preferences and therefore bypasses both approximations.

**Results**    In Table 2, we fine-tune a Mistral-7B model on UltraFeedback and compare SPO/SPO-abs against the baselines. Results show that our methods outperform the recent baselines for preference optimization. This improvement can be attributed to SPO/SPO-abs's ability to exploit all information in the reward dataset. Among the two versions of our method, SPO-abs performs better than SPO, showing the importance of maintaining the absolute values for the best responses' rewards.

## 5.2   EXPRIMENTS ON THE PREFERENCE DATASET

The previous experiments focus is on the reward dataset with $K > 2$ responses per input prompt $(x \rightarrow \{y_i, r_i\}_{1:K})$. However, at present most alignment datasets are pairwise $(x \rightarrow \{y_w, y_l\})$, making it essential also to evaluate our proposed methods in pairwise preference settings. Since SPO is equivalent to DPO when only pairwise preference data is available, we will focus on comparing and clarifying the differences between the DPO and SPO-abs algorithms.

Table 3: Comparison between SPO-abs and DPO in LLM alignment. We focus on the reasoning capabilities on the complex tasks. The highest number in each benchmark is **bolded**. We mark numbers that have decreased ($\downarrow$) after training.

|  | Model | Mixtral-7B | | |
|---|---|---|---|---|
|  |  | SFT | + DPO | + SPO-abs |
| Reasoning | BBH | 60.8 | 61.1 | **61.2** |
| Coding | HumanEval | 27.8 | **30.8** | 30.3 |
|  | LeetCode | **3.3** | 2.4 ($\downarrow$) | 3.2 |
| Math | GSM-Plus | 28.3 | 19.4 ($\downarrow$) | 30.4 |
|  | MATH | 5.8 | 6.4 | **9.8** |
|  | TheoremQA | 7.1 | **8.9** | 8.8 |
|  | SVAMP | 26.9 | 34.1 | **40.8** |
|  | ASDiv | 40.8 | 46.1 | **52.6** |

**Preferecne dataset** We conduct experiments on the UltraInteract dataset Yuan et al. (2024b). ULTRAINTERACT is a high-quality dataset aimed at enhancing the reasoning abilities of large language models through complex tasks. It supports both supervised fine-tuning and preference learning by providing a preference tree for each instruction, including reasoning chains with diverse strategies, multi-turn interactions, and paired correct and incorrect data. The dataset was curated with an emphasis on complexity, quality, and diversity, focusing on math problem-solving, code generation, and logical reasoning. It includes ground-truth solutions to ensure high-quality oversight signals. We consider fine-tuning Mistral-7B and Mistral-8×7B models on UltraInteract. Before alignment, we perform SFT on UltraInteract's preferred responses.

**Evaluation datasets** We evaluate the model's performance in various challenging tasks. This includes:

- BBH (Suzgun et al., 2022). It is a subset of the BIG-Bench, a diverse evaluation suite for language models. BBH focuses on a suite of 23 challenging tasks from BIG-Bench that were found to be beyond the capabilities of current language models. These tasks are ones where prior language model evaluations did not outperform the average human-rater. The BBH tasks require multi-step reasoning, and it was found that few-shot prompting without Chain-of-Thought (CoT), as done in the BIG-Bench evaluations, substantially underestimates the best performance and capabilities of language models. When CoT prompting was applied to BBH tasks, it enabled PaLM to surpass the average human-rater performance on 10 of the 23 tasks, and Codex to surpass the average human-rater performance on 17 of the 23 tasks.

- HumanEval Chen et al. (2021). It used to measure functional correctness for synthesizing programs from docstrings. It consists of 164 original programming problems, assessing language comprehension, algorithms, and simple mathematics, with some comparable to simple software interview questions.

- LeetCode Guo et al. (2024). LeetCode presents competition-level problems, offering significant challenges that test the model's problem understanding and code generation skills.

- Math tasks. In this work, we include 5 challenging tasks for math problem solving: (a) GSM-Plus Li et al. (2024). (b) MATH (Hendrycks et al., 2021). (c) TheoremQA (Chen et al., 2023). (d) SVAMP (Naeem et al., 2014). (e) ASDiv (Miao et al., 2021).

**Evaluation metrics**

**Results** Results are presented in Table 3. Overall, SPO-abs consistently outperforms DPO in various benchmarks. Notably, we observe DPO hurts the overall performance in some of the reasoning tasks compared to the SFT models. This indicates that DPO might not be suitable for improving reasoning abilities, which echoes findings in concurrent work Yuan et al. (2024b). In contrast, SPO-abs shows clear improvement on the 7B models.

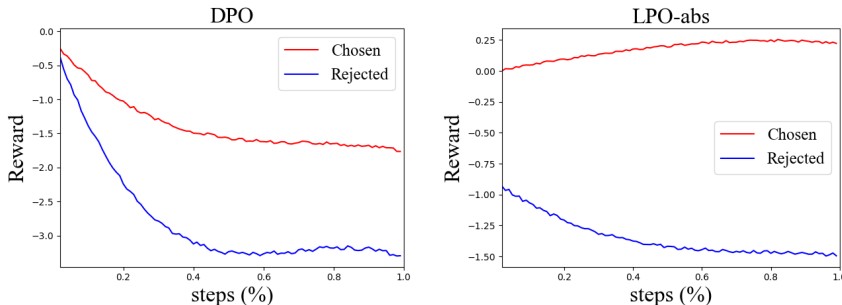

Figure 1: Comparision of data likelihood between DPO and SPO-abs, on the preference dataset UltraInteract.

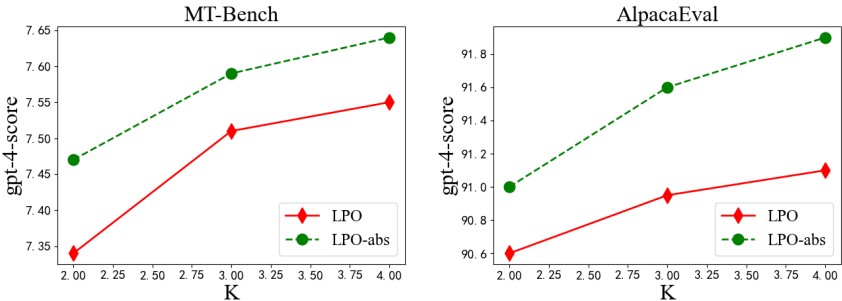

Figure 2: Evaluation results under different numbers of responses.

### 5.3 FURTHER ANALYSIS

**Visualization of likelihood for the chosen and rejected** To demonstrate the distinct optimization characteristics that cause performance differences between SPO-abs and DPO, we inspect how the data likelihood changes during training. The results are shown in Figure 1. As shown in Figure 1. The likelihood of preferred responses interestingly decreases after DPO training and increases for SPO-abs training. The decreasing chosen likelihood trend is concerning because it directly contradicts the maximum-likelihood objective used during the SFT stage. This drawback is exacerbated in reasoning tasks, where the preferred response is often the ground truth answer. Consequently, we hypothesize that SPO-abs' superior performance in reasoning tasks is due to its ability to avoid decreasing chosen likelihood. Since DPO is essentially a specialization of SPO, their contrasting likelihood trends can be explained theoretically. SPO-abs adjusts the absolute likelihood of data, while DPO or SPO only considers relative likelihood across different responses. Thus, a declining chosen likelihood directly contradicts SPO-abs's training objective but not DPO's.

**Effects of the number of responses** $K$ Figure 2 plots how different values of $K$ affects the LLM alignment performance. From Figure 2, we observe consistent performance improvements when increasing the number of data responses from $K = 2$ to $K = 4$ for both SPO and SPO-abs algorithms. In addition, we observe the SPO-abs outperforms SPO under different settings, demonstrating the importance of maintaining the absolute values for the best response's reward.

Figure 2 also conveys an important take-away. Previous practices always ensure selecting the highest performing response when constructing preference data. The assumption behind this strategy is that the dataset's best-performing response determines the upper limit of alignment performance. However, our experiments contradict this assumption. Results in Figure 2 indicate that extra sub-optimal responses can also be advantageous for policy training.

## 6 CONCLUSION

In this work, we thoroughly investigate the language model alignment problem in the context of explicit reward settings. By drawing inspirations from knowledge distillation, we introduce two novel and practical algorithms: SPO and SPO-abs. Our proposed methods are uniquely suited for both reward data and preference data, including DPO as a special case. Our experiments show that our reward-based alignment methods, SPO/SPO-abs, outperform the preference baselines like DPO and KTO by fully leveraging sub-optimal responses in reward datasets. In preference settings, the pairwise version of SPO-abs outperforms DPO in complex reasoning tasks by effectively preventing data likelihood from decreasing.

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
