# OpenReview forum: "A Novel Soft Alignment Approach for Language Models with Explicit Listwise Rewards"
_ICLR.cc/2025/Conference — ICLR 2025 Conference Withdrawn Submission_

### Official Review · Reviewer_sF5D · 2024-10-24

**Soundness:** 1
**Presentation:** 1
**Contribution:** 1
**Rating:** 1
**Confidence:** 5

**Summary:**

**This paper is almost the same as [1].**

Some Evidence:
- The "Preliminary" section in this submission copies the "Background" in [1]
- The "Method" section (section 4) is exactly the same as section 3 in [1], with the same derivation and objective.
- Table 1 in this submission is almost the same as Table 1 in [1].
- Baseline results in Table 2 in this submission is exactly the same number as in Table 2 in [1].

**This is plagiarism and should be desk rejected.**

Reference:
[1] Noise Contrastive Alignment of Language Models with Explicit Rewards. Chen et al. NeurIPS 2024. https://arxiv.org/abs/2402.05369.

**Strengths:**

N/A

**Weaknesses:**

N/A

**Questions:**

N/A

**Details Of Ethics Concerns:**

Please see summary.

---

### Official Review · Reviewer_hCh9 · 2024-10-31

**Soundness:** 2
**Presentation:** 1
**Contribution:** 2
**Rating:** 3
**Confidence:** 4

**Summary:**

This paper proposes two methods SPO and SPO-abs that are specifically designed to learn from both preference as well as scalar reward datasets. They conduct experiments with SPO policies learning from both reward and preference datasets and evaluate on benchmarks like MT-Bench, AlpacaEval and Ultrainteract. Their experiments suggests that their methods surpass previous preference learning baselines like DPO as well as pointwise learning methods like KTO on many of these benchmarks.

**Strengths:**

The paper clearly specifies its goal to devise an algorithm that learns from both preference labels as well as pointwise scalar rewards, that are both typically found in popular preference alignment from human feedback datasets. The benchmarks and baselines chosen are relevant and reasonable.

**Weaknesses:**

The paper has some crucial weaknesses (unless I am missing something): their motivation for choosing a soft-label based knowledge-distillation (KD) approach is not clear and their related work does not mention any KD-based or Plackett-Luce based works that their main contribution (eq. 9) is clearly based off of. Furthermore, there are non-trivial presentation issues, many typos, mislabelling in figures and inconsistencies in prose vs figures etc in the current manuscript. Please read the questions for more weaknesses.

**Questions:**

I find the initial related work to be too broad without sufficient coverage of recent work in offline alignment. TBH, both the first two paragraphs in related work seem too generic and unlike typical alignment papers. While this is not a problem in itself, the paper also does not sufficiently address more recent work in the direct-alignment/RLHF space and does not allude to any works in knowledge distillation. I also have concerns about claims that  algorithms like KTO are only applicable to pairwise preference data since KTO clearly applies to pointwise preferences as a valid data point.  I request the authors to provide more clarification in this: as this will help motivate their scalar reward learning with KD formulation and make it much clearer for the reader.


Line 154: “Compared with constructing preference datasets, annotating each response with scalar rewards can
be more flexible and convenient.” —> are there any citations to back this claim? As far as preference annotations by humans of LLM responses are concerned, it is intuitively easier to get preferences/choices given as pair than to get exact scalar rewards for responses. This is because getting preferences only depends on the pair in focus while the annotator has to calibrate wrt the data distribution in assigning scalar rewards [1].


Line 469: The fig.1 plot seems to suggest expected rewards are plotted against steps and the numbers of the Y-axis are named as *rewards* in the legend. However, the prose in the paper claims y-axis represents likelihoods of the chosen and rejected responses. Are these rewards (as defined in the paper as log-ratios with the baseline policy) or average likelihoods of chosen and rejected responses?  What is also concerning is that both fig.1 and 2 has typos in legends: SPO-abs is written as LPO> Can the authors provide some clarification on these issues and also for how many steps were these DPO and SPO-abs models trained since the x-axis in fig.1 represent percentage of total steps which does not clarify the total steps used for training.


Presentation issues/typos;

—“In order to
draws latent knowledge from the these powerful LLMs..”—> In order to draw..
—Line 323: “We exam the following baseline methods:”—> We examine …

—Line 467: “As shown in Figure 1. The likelihood of preferred responses” —> should be a comma with no caps on “the” likelihood…

—Line 395: “Preferecne dataset” —> Preference dataset

---

### Official Review · Reviewer_vBuG · 2024-11-01

**Soundness:** 2
**Presentation:** 2
**Contribution:** 2
**Rating:** 3
**Confidence:** 3

**Summary:**

Broadly, this paper studies the problem of alignment with offline algorithms such as DPO. The paper is distinguished from prior work in two ways:
* While most prior work has focused on the setting where only pairwise preference judgements are available, this paper focused on the setting with scalar rewards assigned to a set of generations.
* The authors propose new algorithms based on DPO, termed SPO and SPO-abs.

SPO can be seen as a generalization of DPO, with two modifications:
* The objective considers >2 generations per prompt, generalizing from the binary (i.e. K=2) case to the multiclass (K>2) case.
* The prediction target is “soft” (i.e. based on the distribution from some teacher model) as opposed to “hard” (i.e. one-hot label from preference dataset).

SPO-abs adds an additional term to the objective function that incentivizes assigning higher likelihood to preferred generations.

The authors compare SPO and SPO-abs with DPO and other baselines across several settings.

**Strengths:**

* The paper studies a generalization of the DPO objective to the multiclass setting, which could be a more efficient way to train models when there are multiple generations per prompt.
* The paper evaluates the performance of the proposed objectives across multiple settings.

**Weaknesses:**

I think this paper could provide some interesting insights with some modifications. However, I think there are some serious weaknesses with the current version. To recap, the proposed algorithms vary from DPO in three ways:

1. They generalize DPO from the binary to multiclass setting.
2. They use a soft distribution from a teacher model as opposed to the “one-hot” labeled distribution.
3. The `-abs` variant includes an additional term aimed at regularizing towards higher likelihood for preferred responses.

I have concerns about each of these contributions individually.

1. The generalization of DPO to the multiclass setting seems to be the most interesting contribution, as I am not aware of prior work studying this. However, it does not seem like this contribution was evaluated sufficiently on its own. Setting aside the other differences between SPO and DPO (i.e. “hard” vs. “soft” labels), when multiple generations per prompt are available, should we use the proposed multiclass objective or a binary objective over pairs of generations? What are the pros and cons? This is a very interesting question, but the DPO-1vs0 and DPO-pw baselines seem to conflate the other differences in SPO vs. DPO. It would also be good to consider the computational efficiency of such an approach. Is it more memory intensive to have an objective over K>2 generations?

2. The use of a soft target distribution from a teacher vs. a hard target distribution has been proposed by prior work, which is not discussed here, e.g. the “Distilled DPO” (d-DPO) objective of Fisch et al. 2024 (https://arxiv.org/abs/2405.19316). I have not checked the math rigorously, but the proposed objectives seem to be equivalent for the K=2 case. It is still interesting to study a generalization of this objective to the K>2 case, but prior work should be discussed, and it feels misleading for the paper’s title to stress the *novelty* the proposed methods. The comparison between “hard” and “soft” labels also seems to be confounded by the fact that the “teacher model” is so much larger and more powerful than the “student” model being used as the policy. If we have the resources to train such a large “teacher” model, why not train an equally large “student”?

3. For the additional objective in SPO-abs vs. SPO, this also seems to be lacking contextualization in prior work. For example, the authors say “We hypothesize the main cause of this decreasing likelihood is that SPO methods only adjust relative rewards among responses, rather than optimizing their absolute value.”, but this is more or less exactly the hypothesis proposed and studied by prior work (e.g. Pal et al. 2024 (https://arxiv.org/abs/2402.13228)). The proposed new term in the SPO-abs loss did not seem well motivated, i.e. why choose this specific formulation vs. some other? There was a mention of a connection to NCE, but this seemed underdeveloped and the connection was not clear to me. And, more importantly, it’s not clear why some approach from prior work, e.g. based on “DPO-Positive” from Pal et al. 2024 would not be sufficient? Minimally, this should be compared with empirically. Finally, some claims related to SPO-abs seemed confusing, e.g. the authors state “SPO-abs can also guarantee convergence to the optimal LM policy” but it’s not clear what guarantees are offered, or what evidence is provided to support such guarantees.

Therefore, I think the paper would greatly benefit from a revision that more clearly establishes the connection to prior work and experiments that better disentangle the impact of the various aspects of the proposed methods. Proper contextualization with prior work and understanding the impact of the individual contributions is especially important given how crowded the space of proposed DPO variants has become.

While some reviewers may take issue with the focus solely on the offline setting (and not comparing with online methods) or the limited model scales explored in the experiments, these seem like reasonable choices to me given the expense and complexity of experiments in this area.

**Questions:**

Please see weaknesses above.

There are also many small grammatical errors throughout the paper. While most of these do not significantly affect readability, they may be worth addressing, e.g.:
* Abstract: “a adaptive loss” -> “an adaptive loss”
* Abstract: “that inject listwise” -> “that injects listwise”

Other nits:
* Should use \citep vs. \citet in several places, e.g. lines 43 and 47
* “dark knowledge“ seems like an odd way to describe the information contained in the knowledge distillation loss
* Table 2 - the underline indicating the second highest number appears to be in the wrong place for the AlpacaEval column

---

### Official Review · Reviewer_1Y5F · 2024-11-04

**Soundness:** 2
**Presentation:** 3
**Contribution:** 2
**Rating:** 3
**Confidence:** 4

**Summary:**

This paper proposes a new offline alignment method - SPO and SPO-abs (I have seen SPO already at least twice, here is one instance https://arxiv.org/abs/2401.04056 so maybe a different acronym is needed).

SPO is different from DPO is a few ways:
1. Extension from binary classfication from multiclass classifiction by changing the sigmoid cross entropy loss in DPO to softmax cross entropy in SPO. This

2. Adding softness by assuming a teacher model provides a distribution over the multiple possible responses and minimizing cross entropy with respect to that distribution (instead of assuming only a single response is labeled as gold). Notice this strongly assumes we can get a scalar reward for each response, which seems problematic unless the annotator is a machine learning model and not humans (see weaknesses below)

3. in SPO-abs - adding a term that interprets rewards as logits of a sigmoid and essentially tries to maximize the log probability of samples from the base model. This is meant to combat the problem of decreasing probability in DPO and is somewhat orthogonal to the other points made.

Results show that when using ultrafeedback as a reward/preference dataset and evaluating on MT-Bench and Alpaca-eval one obtains some gains compared to DPO and some natural extensions.

**Strengths:**

* Some of the problems identified in this work seem convincing -- the use of only two outputs, and the problem of decreasing likelihood of preferred responses.

* Results show improvement compared to baselines on mt-bench and alpaca-eval

* Analyasis shows reguarlization in SPO-abs in effective, indeed the prob. of the preferred response does not decrease anymore.

**Weaknesses:**

* The authors claim that using a list of rewards is more efficient and convenient than using preferences. This runs in counter to past work even from the original summarization paper by Stiennon et al. (learing to summarize...) where the motivation to get preferences is that it is very hard to get scalar rewards from humans. And also counter to many other works (for example, self-play preference optimization, which is also coined SPO, where they show that scalar rewards are problematic because human preferences do not obey transitivity. So most empirical evidence points against using scalar rewards. The only reason to do this seems to be if you just have a model from which you distill the reward which is what happens in this work -- but the authors don't talk about this or argue for this. Is there claim that this approach is good assuming you are anyway going to use a teacher model to learn from? If the method is mostly applicable for model teachers that would be good to state precisely and as a limitation

* The authors seem to not acknowledge/know relevant past work.
(a) Using soft labels instead of hard labels - the paper "robust preference optimization" by fisch et al. from about 5 months ago already discusses at length use of soft labels. I think simple use of soft instead of hard lables was done even earlier in "Human Alignment of Large Language Models through Online Preference Optimisation" but I am not 100% sure.
(b) There have been a few papers already addressing the problem of reducing likelihood - one is Pal et al. that the authors do cite but don't really mention the fact that they have a positive-DPO variant that adds a similar term for regularization as well as the Fisch et al. paper from above as well as Liu et al from 5 months ago (your SFT loss is implicity an adversarial regularizer)
(c) Googling for a minute I found work on using lists of generations in alignment - LIRE -- https://arxiv.org/pdf/2405.13516

* The extension of the binary case to multiclass (when you don't consider softness) is somewhat incremental. Moreover, without softness it doesn't really exploit the full information in the list of generations - it only maximizes the probability of the single preferred response but doesn't take into account the relative preference of generations that are not the top-ranked ones. In assistant setting it is very hard to assume there is a single gold response, and thus modeling this as multiclass where there is a single correct class seems like a problematic modeling assumption.

* The statement of what the authors are trying to solve is unclear - is it addressing multiple responses? is it addressing the case with scalar rewards? is it just the conjunction of both? is it the likelihood ratio decrease of DPO? It is hard to understand what the authors view as the key contribution.

* Related work - the first paragraph in my humble opinion distracts from the flow - we don't need to go all they way back to BERT for this paper.

* Experimentally - I did not understand the motivation for choosing the reasoning setup - is there any reason to think that SPO will be good for this setup? Is this an arbitrary choice? Also, there is a mismatch between the dataset used for training the the reward model / aligning the model and the actual benchmarks used for evalution and it is hard to reason about the results with the need to also generalize to out-of-distribution settings as part of the experiment.

* minor: "The soft label represents the important dark knowledge from the teacher LLM” I would encourage rephrasing - what does dark knowledge mean?

* minor: The authors use the acronym LPO instead of SPO in the figures in the end, probably by mistake.

**Questions:**

* line 169: what is the index t?

---

### Note · Authors · 2024-11-13

I have read and agree with the venue's withdrawal policy on behalf of myself and my co-authors.